# Application of Nanomaterials in the Prevention, Detection, and Treatment of Methicillin-Resistant *Staphylococcus aureus* (MRSA)

**DOI:** 10.3390/pharmaceutics14040805

**Published:** 2022-04-06

**Authors:** John Hulme

**Affiliations:** Department of BioNano Technology, Gachon University, Seongnam-si 461-701, Korea; johnhulme21@gmail.com; Tel.: +82-31-750-8550

**Keywords:** nanomaterials, vaccines, diagnostics, treatments, biofilm

## Abstract

Due to differences in geographic surveillance systems, chemical sanitization practices, and antibiotic stewardship (AS) implementation employed during the COVID-19 pandemic, many experts have expressed concerns regarding a future surge in global antimicrobial resistance (AMR). A potential beneficiary of these differences is the Gram-positive bacteria MRSA. MRSA is a bacterial pathogen with a high potential for mutational resistance, allowing it to engage various AMR mechanisms circumventing conventional antibiotic therapies and the host’s immune response. Coupled with a lack of novel FDA-approved antibiotics reaching the clinic, the onus is on researchers to develop alternative treatment tools to mitigate against an increase in pathogenic resistance. Mitigation strategies can take the form of synthetic or biomimetic nanomaterials/vesicles employed in vaccines, rapid diagnostics, antibiotic delivery, and nanotherapeutics. This review seeks to discuss the current potential of the aforementioned nanomaterials in detecting and treating MRSA.

## 1. Introduction

Recent projections indicate that by 2030 global antibiotic consumption will have doubled [1], with infections resulting from antimicrobial-resistant (AMR) bacteria expected to claim 10 million lives per annum by 2050. Key to mitigating against such projections is the global implementation of antimicrobial stewardship (AS) and the SENTRY Antimicrobial Surveillance Program, which has successfully reported a decrease in MRSA prevalence since its peak more than a decade ago [2]. However, antimicrobial stewardship remains far from a reality in Africa and India. Studies show that a high proportion of antibiotics used in private and public care settings in African countries are inappropriate [3,4,5,6]. Moreover, even before the pandemic, India faced major AMR challenges, with the prevalence of highly resistant Gram-negative bacteria orders of magnitude higher than many high-income countries. Although the majority of AS interventions currently occur in affluent countries, during the early stages of the pandemic, interventions were relaxed, with a high proportion of COVID-19 patients receiving antimicrobials (pooled prevalence 75%) [7,8], even when confirmed bacterial co-infection prevalence was low (8%). Such exceptional conditions may have contributed to reports of AMR in hospitalized COVID-19 patients. For example, Kampmeier et al. [9] reported *vanB* clones of Enterococcus faecium in COVID-19 subjects from intensive care wards in Germany. In addition, NDM Enterobacterales was also isolated from COVID-19 patients in an Italian teaching hospital prolonging “length of stay” Porretta et al. [10].

Exceptional conditions aside, perhaps most concerning, was the broad application of enhanced chemical sanitization practices and limited UV sterilization procedures employed throughout the entirety of the pandemic [11]. Furthermore, said practices may have resulted in New Delhi Metallo (NDM)-beta-lactamase-producing carbapenem-resistant Enterobacterales isolates being detected in critically ill COVID-19 patients in New York City [12]. In addition, a 2020 study showed the detection rate of *S. aureus* (SA) and MRSA in 180 elderly patients with respiratory tract infection in a psychiatric department in China was higher following increased concentration and frequency of disinfection (Figure 1) [13]. Of the seven MRSA strains detected, antimicrobial susceptibility testing of samples from January 2020 to April 2020 showed that in the absence of a recent epidemiological linkage the increased cases of MRSA infection were most likely attributable to an interactive relationship between microbial disinfectant and antimicrobial resistance. Moreover, the authors suggested future disinfection processes should occur in well ventilated areas in the absence of residents for a prescribed period in order to prevent nasal and pulmonary cavities being exposed to sub-lethal levels of disinfectants if it all possible. Interestingly, genetic disclosure showed newly diagnosed patients were probably exposed to or carrying MRSA as early as 2017–2018, suggesting the application of a rapid diagnostic prior residential admission and workers might be considered. The parallel rise in SA and MRSA cases might also suggest a collective residence (biofilm) and subsequent release of persisters into the sputum. Thus, following decolonization treatments, a rapid diagnostic for quorum and other film markers (recurrent risk) or a change in the residential mouth washing regime might be an option going forward.

MRSA is renowned for its ability to acquire resistance to front-line treatment options as typified by vancomycin-intermediate *S. aureus* (VISA), heterogeneous VISA(h-VISA), and vancomycin-resistant *S. aureus* (VRSA). MRSA resistance to vancomycin is acquired via the transfer of the van gene clusters (*vanA* and *vanB*), which provide resistance by altering the drug target from D-alanine-D-alanine to D-alanine-D-lactate [14,15]. Other types of resistance involve the transfer of plasmid-mediated resistance genes (*vanA*, *vanB*, *vanD*, *vanE*, *vanF*, and *vanG*) from vancomycin-resistant Enterococcus faecium (VRE) or *Clostridium difficile* (CD) [16]. In addition, VRSA tends to be multidrug-resistant (MDR) against a diversity of currently available antibiotics, including β-lactams [17]. Moreover, a recent report showed that vancomycin-resistant isolates are >250 times less susceptible to narrow-spectrum fidaxomicin compared to fidaxomicin-sensitive strains, even though these two antibiotics have different mechanisms of action [18], suggesting narrow-spectrum antibiotics (NSA) should be prioritized as first-line treatments when possible. Furthermore, recent studies show VRSA frequency increased threefold from 2006 to 2014, and 1.2-fold between 2006 and 2014 and between 2015 and 2020 [19]. 

Efforts to reduce dependency on vancomycin by combining it with b-lactams and daptomycin has showed promising results. However, such combinations can result in a higher incidence of nephrotoxicity [20]. Moreover, traditional therapies often fail to reach suitable intracellular levels in bacteria and phagocytic hosts. An alternative approach involving nanomaterials via enhanced diagnostics and drug encapsulation have sought to enhance drug efficacy whilst reducing acute toxicity in the host. 

Therefore, many researchers have focused on incorporating nanomaterials with rapid diagnostics and efficient drug delivery systems to meet the challenge of broad-spectrum antibiotic resistance encountered with traditional therapies. These vehicles can be composed of biomimetic membranes, liposomes, polymers, chitosan, and inorganic materials. Numerous studies have shown that many of these materials are compatible with and enhance the sensitivities of traditional laboratory and point of care diagnostics [21]. The integration of these compatible nanomaterials is so refined that multiplexable autonomous disposable nucleic acid amplification tests (MAD NAAT) constructed on 2D paper networks can detect MRSA in less than 1 h [22]. 

Antibiotics delivered via these nanomaterials benefit from reduced enzyme deactivation and improved efficacy. Moreover, if the material itself induces antimicrobial activity via reactive oxygen species independent pathways, the potential for resistance can be reduced. Other advantages include extended retention time, improved serum stability, reduced hepatotoxicity, and gut microbiome perturbation [23].

In addition, these carriers can act as decoys, reducing the impact of virulent microbial factors such as toxins, adhesions, and secretory systems, thereby minimizing disruption to indigenous microflora. The advances come at a time when the effect of subinhibitory antibiotic concentrations on outer membrane vesicle production and the potential for the dissemination of resistant genes from susceptible bacteria is becoming apparent [24]. The complex bi-directional role of extracellular vesicles in infection and antibiotic resistance is beyond the scope of this review. Kim and He et al.’s studies are recommended for those readers seeking further insight regarding extracellular vesicle (EV) production and their roles in vancomycin and methicillin-induced biofilm formation [25,26]. This review discusses five areas where natural and synthetic delivery carriers/vehicles are used to combat MRSA. These areas include (1) vaccines, (2) rapid diagnostics, (3) antibiotic delivery, (4) nano-stealth coatings, and (5) biofilm inhibition. Advances in these areas bring us ever closer to tailored antibacterial therapies that respond to changes in *S. aureus* susceptibility, virulence factors, host organism infiltration, and colonization resistance.

## 2. Vaccines and Nanovesicles

Vaccines can reduce the spread of antibiotic-resistant pathogens, antibiotic usage, and the risk of symptomatic disease and associated costs. Recent predictions suggest that vaccines could play a significant role in controlling antibiotic resistance [27]. However, the Gram-positive pathogens [28] *Clostridium difficile* (CD), MRSA, and SA have a wide array of virulent determinants at their disposal, including surface proteins [29], glycopolymers [30], and multiple secreted proteins, such as superantigens (T cell impairment), hemolysins, proteases, and toxins [31], allowing them to circumvent and impair the hosts innate and adaptive immune response, reducing vaccine efficacy. Despite promising preclinical results, *S. aureus* monoclonal and polyclonal vaccines targeting major toxin (a-hemolysin (Hla), Panton-Valentine leukocidin (PVL), and phenol-soluble modulins (PSMs)) failed clinical trials [32,33], suggesting specific antibodies were insufficient to prevent pathogenic escape. Coupled with the recent withdrawals of the StaphVAX (bivalent polysaccharide and protein conjugate vaccine) developed by Nabi Biopharmaceuticals, V710, a vaccine trialled by Merck [34], and the four-antigen vaccine candidate SA4ag composed of capsular polysaccharide conjugates and recombinant proteins from Pfizer [35], there is an urgent need to develop additional vaccine candidates akin to virulence factor SpA and the pore-forming toxins leukocidins as well as novel adjuvants currently in the preclinical phase of development [36]. However, the cost of developing a multicomponent vaccine currently outweighs the economic benefits. Therefore, researchers have sought cheaper and naturally available alternative platforms for vaccine development.

### The Role of Gram-Negative and Positive Extracellular Vesicles in Vaccine Development

EV formation by Gram-negative bacteria was first observed by electron microscopy more than fifty years ago [37], and these bacteria secreted what is now referred to as outer membrane vesicles. Since then, OMVs have emerged as commercially promising vaccine platforms suitable for human use [38]. Ranging in size from 20 to 300 nm, OMVs are vesicles principally composed of a lipid bilayer, on and within which proteins, lipoproteins, peptidoglycans, DNA, RNA, and various multiple pathogen-associated molecular patterns (PAMPs), including lipopolysaccharide (LPS), are housed (Figure 2) [39]. OMV’s versatility has led to its employment in various applications, including adjuvant and vaccine synthesis, antibacterial treatments, and bioimaging [40].

Until recently, Gram-positive EV biogenesis and its contents remained poorly understood. Numerous studies have since characterized the protein content (or cargo) and interaction of *S. aureus* EVs with eukaryotic host cells during infections [40,41]. For instance, *S. aureus* vesicles are important in the development of atopic dermatitis (AD), a chronic inflammatory skin disease [42,43]. EVs containing the pore-forming toxin α-hemolysin increased necrosis and AD-like skin inflammation in mice compared to mice exposed to soluble α-hemolysin [44]. Moreover, the complete cascade through which *S. aureus* EVs activate the inflammasome in macrophages showed that EVs function as an efficient virulence factor delivery system [45]. Finally, the EV core proteome has been deduced by comparing EVs from different *S. aureus* isolates (both human and animal) [46].

EV and OMV formation are considered an essential process involving several factors influencing stress responses and specific gene expression [47]. In the lab, EV (SA) production is initiated by growing then harvesting (growth and stationary phase) bacterial cultures in the presence of a sub-inhibitory concentration of antibiotics such as vancomycin [48] and when mimicking infection stress in the absence of a metal ion (usually iron) or ethanol [49]. In a recent study, Kim et al. investigated whether EVs from MRSA under stress conditions or normal conditions could reduce the susceptibility of bacteria in the presence of several β-lactam antibiotics. EVs harvested from MRSA cultures under antibiotic (ampicillin)-stressed conditions provided a 22.4-fold reduction in antibiotic susceptibility compared to unstressed EVs. EVs secreted from ampicillin-stressed MRSA afforded some protection to several species of Gram-negative bacteria, including Escherichia coli and Salmonella spp. Proteins related to the degradation of β-lactam antibiotics were abundant in ampicillin-induced EVs [25]. Similarly, EVs harvested from MRSA (USA 300 strain ATCC BAA-1717) grown in sub-therapeutic concentrations (0.5 mg/L) of Van [50], doubled the Van MIC for MRSA. Furthermore, the presence of EVs increased survival of MRSA pre-treated with sub-MIC concentrations of Van in whole blood and upon exposure to human neutrophils but not in human serum. In another study, Wang et al. [51] employed penicillin G (PenG) to increase the EV yield from JE2, a *S. aureus* USA300 strain representative of the prevalent US CA-MRSA clone. Using mutated JE2, in which protein A and the toxins Hla, Panton-Valentine leukocidin (Luk-PVL), LukED, HlgCB, SelX, and PSMs expression were suppressed, the authors showed the resultant EVs to be non-toxic to mammalian cells and capable of eliciting cytolysin-neutralizing antibodies, protecting the animals in a lethal sepsis model, indicating that these naturally produced vesicles have potential as a novel vaccine platform.

The ability of temperature to modulate antibiotic resistance has been known for decades, requiring localized photodynamic therapies (PTT) to exceed > 50°C in order to minimize the dissemination of resistant genes. Consequently, the effects of lower temperature on EV production have been overlooked. However, in a recent study, Briaud et al. [52] demonstrated the importance of lower temperature in vesicle production and packaging. At high temperatures 40 °C, packaging of virulence factors and protein and lipid concentration increased with a reduction in the overall RNA abundance and protein diversity. In contrast, the EVs secreted at 34 °C were more cytotoxic toward THP-1 cells(macrophages), and the EV proteome was more diverse. These results suggest that vesicle content can be modulated by applying small changes in ambient temperatures (Temp and UV).

## 3. Multiple Roles of Nanomaterials in Rapid MRSA Diagnostics

Rapid, cost-effective identification of causative pathogens and determination of their antibiotic resistance profiles should ideally precede initiation of therapy [53]. The first stage in MRSA identification (inoculation and blood cultures) can take from 18 to 48 h, depending on the sample volume and quality, which may be too long for critically ill patients who require administration of a specific antibiotic therapy within 24 h after the onset of sepsis [54,55]. To date, methicillin resistance (MR) SA strains, such as hospital-acquired (HA)-MRSA and community-acquired (CA)-MRSA, represent the most serious challenge to public health [56]. Genotypic identification relies on detecting SA-specific genes, such as *spa*, *nuc*, and *fem*, combined with the *mecA* gene [57]. The *mecA* gene codes for the penicillin-binding protein (PBP2a) and is carried by the staphylococcal cassette micro chromosome (SCCmec), a mobile genetic element [58]. Fourteen types (I-XIV) of SCC elements have been reported, all carrying the *mec* and cassette chromosome recombinases (CCR) gene complexes [59]. CA-MRSA can be distinguished from HA-MRSA by the presence of SCCmec types IV and V and the Panton-Valentine Leukocidin (PVL) exotoxin, the latter often associated with necrotizing pneumonia and severe skin infections [60].

In the last two decades, immunomagnetic magnetic nanoparticles (MNPs), particularly superparamagnetic nanoparticles (SPMNPs), have attracted a lot of commercial and academic attention due to their excellent magnetic properties, low cost, assay versatility, and higher capture efficiency [61].

In addition to sample preparation, SPMNPs (e.g., Fe_3_O_4_-Ag, FeO_4_-Au, and FePt-Ag) can be used directly or as part of a multifunctional composite to improve the sensitivity of optical and electrochemical immunoassays. The unique chemical properties of noble metal NPs, particularly AuNPs, render them compatible with various optical and electrochemical methods such as UV spectroscopy, colourimetry, fluorimetry, and electrochemical impedance spectroscopy (EIS) [62]. In 2017, Kearns et al. combined lectin-functionalized silver-coated MNPs with optically active antibody-coated silver NPs to isolate and detect three bacterial pathogens, including MRSA, in an Eppendorf tube using surface-enhanced Raman spectroscopy (SERS) [63]. Li et al. differentiated MRSA from MSSA isolates in blood samples by magnetic separation and SERS in several stages. Firstly, polyethyleneimine-modified magnetic microspheres (Fe_3_O_4_@PEI) were used to capture bacteria directly on blood samples. Following 20 min of incubation with Fe_3_O_4_@PEI, the complex Fe_3_O_4_@PEI–S. aureus (magnetically isolated bacteria) was plated on agar with and without antibiotics and incubated overnight. Then, using SERS fingerprints from a single colony, 11 MSSA and 13 MRSA were correctly identified by analyzing their Raman signature regarding lipids, amino acids, and nucleic acid content [64]. The outstanding capture efficiency of streptavidin–magnetic beads was also utilized by Potluri et al. in the simultaneous detection of *mecA* and *femA* genes by surface-enhanced Raman spectroscopy. The authors’ SERS–PCR system successfully quantified *mecA* and *femA* in 14 MRSA clinical samples and four non-staphylococcal species in Eppendorf tubes [65]. Silver nanoparticles (AgNPs) are routinely employed in bacteria detection, but their negative surface limits SERS applications. Recently, Chen et al. reported a novel SERS method using positively charged AgNPs (AgNPs+) to rapidly identify MRSA [66]. Employment of AgNPs+ enabled superior SERS enhancement, which provided higher-quality and reproducible SERS fingerprinting spectra. Researchers subsequently identified differences in DNA, lipids, and protein spectra for MSSA and MRSA cell membranes. These differences allowed the researchers to distinguish MSSA (52 strains) and MRSA (215 strains) from clinical samples using partial least squares discriminant analysis (PLS-DA). The advantages of combining optical and electrochemical techniques were also explored by Lv et al. [67], in which a doxorubicin (DOX) probe and a nanostructured Au-modified indium tin oxide electrode surface were used to simultaneously measure the SERS and EIS of multidrug-resistant MDR SA (MDR-SA) in pure and contaminated milk. The combined approach exhibited an LOD of 1.5 × 10^2^ CFU/mL of MDR-SA in real samples.

The aggregation of NPs induces interparticle surface plasmon coupling, resulting in a blue shift in the visible absorbance spectrum. This colorimetric change has been utilized to detect bacteria-specific DNAs, proteins, and live cells. For example, as early as 2004, Storhoff et al. used AuNPs to detect the *mecA* gene in MRSA genomic DNA samples [68]. The approach was effective in discriminating MRSA from methicillin-sensitive *S. aureus* strains. More recently, Chan et al. also used AuNPs for direct colorimetric PCR detection of MRSA in 72 clinical specimens; the performance was comparable with real-time PCR assays but at a lower cost per reaction [69]. The cost per reaction can be reduced further if the colourimetric *mecA*-based PCR qualitative test is conducted in an Eppendorf tube or on a paper substrate. For example, Eldin and the group carried out the specific detection of the *mecA* gene using AuNPs conjugated with complementary ssDNA strands in an Eppendorf tube [70]. This method produced visible colour changes, which was confirmed using UV spectroscopy and provided high sensitivity of 90.9% at 10 μL of DNA target per 200 μL of the total volume of the reaction mixture.

Qualitative colourimetric identification of pathogenic bacteria utilizing Eppendorf tubes or paper substrates by untrained personnel can potentially improve the global surveillance capacity of antimicrobial resistance in a cost-effective manner [71]. With this in mind, a novel paper-based visual sensing platform was fabricated by Zourob and co-workers [72]. The sensing mechanism was based on the proteolytic activity of *S. aureus* proteases on a specific peptide substrate, sandwiched between magnetic nanobeads and a gold surface on top of a paper support. An external magnet was placed on the back of the paper, which promotes the breaking of the peptide–magnetic nanobead complexes. The paper-based method was an inexpensive technique with high sensitivity capable of visual detection of MRSA. Another novel point of care device called Clear Read, a customized colorimetric assay for detecting DNA molecules without any amplification, was developed by Ramakrishnan et al. to detect the *mecA* gene in clinical samples. The procedure involved oligonucleotides bound to a solid matrix conjugated with AuNPs. The AuNPs were catalytically coated with silver, resulting in a six-fold increase in the output signal while requiring only about ~500 ng of DNA to detect target molecules such as the *mecA* gene [73]. With the advent of non-amplification genomic gDNA devices [74] and lateral flow tests employed in the detection of *S. aureus* [75], the application of these tests during the flu season would undoubtedly complement antibiotic stewardship. Failure to detect co or secondary *S. aureus* resultant from flu infection can lead to pulmonary complications [76] (excessive coughing, bilateral fracture), as shown in the computed tomography CT images in Figure 3.

Another novel gDNA assay utilized resistive pulse sensing (RPS), loop-mediated isothermal DNA amplification (LAMP), and AuNPs in the rapid detection of the PVL gene were reported by Kong et al. [77]. Resultant LAMP products called Lamplicons were incubated with two gold nanoparticle probes and modified via biotin-avidin coupling. These coupled particles were put in a tunable nanopore platform (qNano, IZON Science), producing a measurable resistive pulse when the nano-assembly passed through the pore. The resulting LOD for detecting MRSA DNA template was as low as 530 copies, with the quantitative process completed within 2 h. This approach utilizes a straightforward and sensitive protocol requiring one single temperature and four primers to isolate and amplify target DNAs by LAMP. Results demonstrated that the combined LAMP-based AuNP RPS was an effective tool for distinguishing CA-MRSA from nosocomial MRSA. Furthermore, Lee and colleagues made a microfluidics-based diagnostic assay with sensing probes attached to magnetic beads in the microfluidic channel to detect target DNA from MRSA bacterial strains [78].

Nanostructure (NS) integrated systems incorporating aptamers have been increasingly used in bacterial disease [79,80,81]. Aptamers are small, single-stranded DNAs or RNAs that bind their specific targets with high affinity and selectivity and are produced by systematic evolution of ligands by exponential enrichment (SELEX) or other modified SELEX strategies. Aptamer-functionalized AuNP or gold nanorods (AuNR) solutions were separately added to the MRSA solution containing 10^7^ CFU/mL cells, and each mixture was incubated for 1 h at 37 °C. Under infra-red illumination, Apt@Au NP-MRSA was deemed suitable for MRSA diagnostics, whereas apt@Au NRs was not [82]. Unfortunately, the authors did not test other nanoparticulate geometries such as nano triangles and rings regarding the selectivity of MSSA and h-VISA.

### Graphene Oxide and Fluorescent Nanomaterials

Graphene Oxide (GO) is hydrophilic, and its surface is easily modified with a host of biocompatible polymers such as chitosan, [83] polyethylene glycol (PEG) [84], poly(ε-caproplactone) [85], poly-L-lysine (PLL) [86], and polyvinyl alcohol [87]. Graphene and functionalized graphene have been used effectively in various electrocatalysis and electrochemical biosensing applications, demonstrating significant promise. For example, Wang et al. modified and functionalized a glassy carbon electrode (GCE) with reduced graphene oxide (rGO) and amimopropyltriethoxysilane (APETS) coatings in <2 h. The sensing electrode was prepared by conjugating ssDNA, complementary to target DNA. Electrochemical impedance spectroscopy (EIS) measurements using the sensing electrode demonstrated an LOD of 10–13 M for MRSA DNA [88]. In addition, GO has an incredibly high fluorescence quenching efficiency. Thus, graphene-based nanomaterials can be utilized in the construction of fluorescent transducer-based biosensors. Chen and colleagues used a similar energy transfer method using fluorescent probes and GO to detect the *mecA* gene [89]. The probes consisted of two regions, and one made up of a complementary probe specific for the target gene. The other was a primer responsible for amplifying fluorescent signals after the SYBR Green I dsDNA. The fluorescent emission peaks were recorded at 514 nm for SYBR Green I. FAM also emitted light of the same wavelength, resulting in the amplification of the fluorescent signal. This novel biosensor detected the *mecA* gene with a linear range from 1 to 40 nmol/L and a lower detection limit of 0.5 nmol/L. The lower detection limit of bacteria was 3 × 10^2^ CFU/mL, with a linear range from 10^3^ to 10^7^ CFU/mL.

Similarly, Ning et al. [90] reported a method for the fluorometric determination of MRSA by exploiting target-triggered chain reactions and deoxyribonuclease I (DNase I)-aided target recycling. This experimental bioassay study was carried out using FAM labelled probe with two sections over GO for FRET-based detection of 16 rRNA of MRSA bacterial strain. The FAM-labelled probe adsorbed to the GO by π-stacking, quenching its fluorescence protecting it from DNase I cleavage. After introducing the target sequence, DNA/RNA hybrids resulted, permitting FAM enzyme cleavage to occur, producing a target-induced fluorescence signal. The limit of detection for MRSA 16S rRNA was 0.02 nM. The LOD for bacterial samples was 30 (Colony Forming Unit) CFU mL^−1^ with a linear range from 10^2^ to 10^6^ CFU/mL. A selection of the various nanomaterials used to detect MRSA are summarized in Table 1.

More recently, Liu et al. [91] developed an efficient and versatile method for detecting MRSA DNA sequences in which a nanoparticle-based luminescence resonance energy transfer (LRET) system was utilized. The technique was based on the upconversion of nanoparticles (UCNs) and LRET between NaF4: Yb, Er UCNs, and carboxytetramethylrhodamine (TAMRA), the energy acceptor. MRSA-captured nucleotides were immobilized on the surface of UCNs and released in the vicinity of TAMRA-labelled DNA reporter oligonucleotides. Upon sandwich hybridization, with specific MRSA DNA sequences (Mec-Tar), a shift (543–580 nm) and an increase in the emission wavelength was observed. An LOD of 0.18 nM for MRSA DNA sequences was reported using the UCN-based LRET system. As well as LRET, fluorescent energy transfer (FRET) assays have also been employed to detect *S. aureus*. TNase is a nonspecific endonuclease specifically produced by *S. aureus*. A bacterial count above 10^3^ CFU/g will generate 1 ng/mL of TNase enzyme. Furthermore, Chandan et al. conjugated anti-TNase antibodies to a CdTe QD-streptavidin probe and used them in a simple and inexpensive FRET immunoassay [92]. The assay was successfully validated on naturally contaminated samples, showing good linearity and an LOD of 0.5 ng/mL.

## 4. Antibacterial Agents

### 4.1. Metallic Nanoparticles

The increasing use of MNPs in medicine has led to a growing number of studies exploring the antibacterial mechanisms of MNPs and the potential for resistance [93]. MNPs’ physicochemical properties include their size, shape, charge, zeta potential, surface morphology(roughness), and crystal structure, which are significant elements that regulate the actions of MNPs on bacterial cells. Current research suggests MNPs employ three antibacterial mechanisms: oxidative stress [94], non-oxidative stress [95], and metal ion release [96]. Different MNPs have been used to investigate their efficacy against MRSA, of which the most explored are silver and gold NPs (Ag and Au NPs) [97,98,99,100]. Regarding commercial MNPs applications, AgNPs are the most common, found in cosmetics, nanomedical devices, and food products. Although generally less toxic than silver ions, their ability to induce oxidative stress for a prolonged period in eukaryotic cell lines and subcellular organelles(mitochondria) suggests they could contribute to the early onset of various metabolic diseases (neurodegenerative, cardiac) [101,102,103,104]. The source of this toxicity is open to debate, with many experts suggesting that it is not uncontrolled silver ion release but the shape and size of the particles. Uncontrolled ion release aside, reports can vary regarding AgNP toxicity and the animal models used. Another potential source of toxicity is the solvents employed during particle synthesis. Consequently, many researchers have turned to greener methods, resulting in significant reductions (enhanced particle stability) in geno and cytotoxicity in cell lines, graphene being a notable example [104]. Regarding biomedical usage, lifetime matching, i.e., particle stability to device function, is routinely applied in implants and topical applications in order to minimize toxic events.

Cheaper alternatives to Ag and AuNPs, such as zinc oxide (ZnO) NPs and titanium dioxide (TiO_2_) NPs, have effectively killed MRSA under in vivo and in vitro conditions [105,106]. For example, the application of ZnO NPs to reduce the bacterial burden in MRSA-associated skin infection in murine models has proven effective [107], with one study reporting antibacterial activity of ZnO NPs in MRSA at a concentration of 1875 mg/mL [108]. Similarly, another study reported the bactericidal activity of ZnO NPs with additional insights into the mechanisms of these NPs, which inhibit multiple metabolic pathways, such as amino acid synthesis, in *S. aureus* [109]. TiO_2_ NPs have also been successfully applied with different combinations of antibiotics, such as cephalosporins, glycopeptides, and azalides, showing anti-MRSA activity in a disk diffusion assay. Under UV photoactivation, TiO2 (NPs) form free radicals that lead to their enhanced killing of MRSA [110].

ZnO NPs’ excellent biomedical properties have resulted in their employment in diagnosis, bio-imaging, drug delivery, antimicrobial, and cancer treatments, etc. [111,112]. However, new approaches are needed for ZnO NPs to meet the non-agglomeration requirements of clinical settings. Doping modification is one of the most effective methods to minimize ZnO NPs–bacterial agglomerates. For example, Cu-doped ZnO nanorods exhibit better photocatalytic and antibacterial characteristics than pure ZnO nanorods [113]. Recent work by Khalid et al. [114] tested the antibacterial effects of the Cu-doped ZnO NPs against four bacterial strains, two of which were Gram-positive (*S. aureus*, *S. pyogenes*) and two Gram-negative (*E. coli, K. pneumonia*). Studies showed that Gram-positive microbes were more susceptible to Cu-doped ZnO NPs than Gram-negative microbes. Furthermore, Cu-doped ZnO NPs exhibited better antibacterial activity (than pure ZnO) towards Gram-positive bacteria than Gram-negative bacteria. The same group also investigated the antibacterial activity of cobalt-doped zinc oxide cylindrical microcrystals using similar parameters [115]. The results showed that Co-doped ZnO MCs had better antibacterial activity against Gram-negative bacteria than Gram-positive bacteria.

### 4.2. Liposomes

Due to the ease of formulation, low cost, and compatibility with a plethora of established therapeutic agents, liposomes remain one of the most widely used nano-drug delivery systems. The basic liposome consists of one or more spherical lipid bilayers surrounding an aqueous core incorporating either hydrophilic or hydrophobic compounds [116,117]. The size and the number of layers determine the drug encapsulation efficiency (EE). The circulatory half-life of liposomes is enhanced via pegylation, which improves osmotic stability and inhibits the binding of undesired plasma proteins destined for the reticuloendothelial system (RES) [118]. With the re-emergence of vancomycin-intermediate *S. aureus* (VISA), heterogeneous VISA (h-VISA), and vancomycin-resistant *S. aureus* (VRSA), particularly in Africa, cost-efficient systems that increase the efficacy of vancomycin would be advantageous. Studies involving Van-encapsulated liposomes usually employ the hydration–dehydration or rehydration–dehydration method. For example, in a study to improve the MRSA killing efficiency of Van, Sande et al. [119] prepared two liposomal formulations (Dicethylphosphate (DCP) and dimyristoylphoshatidylglycerol (DMPG)) loaded with Van using the rehydration method. The study reported that both liposomal formulations were approximately two-fold more effective than free-form VAN with minimum inhibitory concentrations (MICs) ranging from 0.3 to 1.25 mg/mL for both liposomes formulations, enhancing the clearance by a magnitude compared to free form Van with minimum bactericidal concentrations (MBCs) ranging from 0.6 to 1.25 mg/mL for both liposomes and 2.5 to 5 mg/mL for free form Van in a systemic murine infection model. Serri et al. [120] investigated the efficacy of a Van-loaded liposomal formulation using conventional lipids, prepared by the lipid film hydration method and evaluated against *S. aureus* and MRSA. The study reported low encapsulation efficiencies (EE), ranging from 0.1% to 9% for the various liposomal formulations. Due to the low EE, the liposomal formulations showed inferior MIC values (3.47 μg/mL) compared to free-form Van (2.4 μg/mL) against *S. aureus* and MRSA (6.95 μg/mL and 4.8 μg/mL), respectively. MBC values also followed a similar trend. Recently, another research group has evaluated Van-loaded conventional liposomes for their antibacterial efficacy against MRSA in an in vivo study. In 2020, Abrishami et al. prepared Van-loaded nanoliposomes using the solvent evaporation method. The study reported the particle size of the liposomal formulation to be 381.93 ± 30.13 nm, having an encapsulation efficiency of 47%. The liposomal formulation was significantly more effective than the freeform vancomycin at each tested time interval (*p* < 0.05). Their results indicated that positively charged and nanosized liposomes showed enhanced therapeutic effects [121].

Novel lipids and pH-responsive lipids have been shown to overcome the acidic micro-environment [122], permitting fusion to the negatively charged cell wall of MRSA at low pH [123]. For example, the work by the Omalo group [124] utilized an advanced nano-drug delivery system composed of oleic acid (OA) and a novel quaternary lipid (QL) to encapsulate Van. Encapsulation efficiencies were 43.06 ± 5.86% and 16.95 ± 1.23% for pH-responsive and non-pH-responsive liposomes. The study revealed that pH-responsive liposomes exhibited better antibacterial activity than free Van at pH 7.4. Results indicated MICs were 2 to 4 times lower for pH-responsive liposomes than Van and non-pH responsive for *S. aureus* (0.98 µg/mL, 3.9 µg/mL, and 1.95 µg/mL, respectively) and MRSA (1.95 µg/mL, 7.8 µg/mL, and 3.9µg/mL, respectively). Moreover, MICs were 8 to 16 times lower at pH 6.0 for pH-responsive liposomes than free Van and non-pH responsive for *S. aureus* (0.488 µg/mL, 3.9 µg/mL, and 1.95 µg/mL, respectively) and for MRSA (0.488 µg/mL, 7.8 µg/mL, and 3.9 µg/mL, respectively). In vivo studies showed that MRSA recovered from mice treated with formulations was 189.67- and 6.33-fold lower than the untreated and bare Van-treated mice. OA-QL liposomes also demonstrated a 1266.67- and 704.33-fold reduction in the intracellular infection for TPH-1 macrophage and HEK293 cells, respectively.

In another study [125], a novel two-chain fatty acid-based lipid (FAL) containing amino acid head groups in the formulation of pH-responsive liposomes for the targeted delivery of vancomycin was reported. The liposomes were characterized by size, surface charge, polydispersity index (PDI), and morphology. In addition, the drug-loading capacity, drug release, cell viability, and in vitro and in vivo efficacy of the formulations were investigated. A sustained drug release profile was observed; SA and MRSA MICs were two- to four-fold times lower for encapsulated Van at pH 7.4 and 6.0 than purified Van. In vivo studies showed similar reductions in MRSA recovered from mice treated with encapsulated Van compared to the control.

Fusogenic liposomes consisting of dioleoyl-phosphatidylethanolamine (DOPE) and cholesterol hemisuccinate (CHEMS) increase the fluidity of the lipid bilayer. Under normal conditions, fusogenic liposomes adopt a liquid crystalline state; however, in the presence of cations, the bi-layer arrangement relaxes, permitting fusion with other membranes. Recent work by Scorboni et al. [126] comparing the in-vitro antimicrobial activity of encapsulated vancomycin in different liposomal formulations against *S. aureus* biofilms showed that vancomycin encapsulated in fusogenic liposomes demonstrated enhanced antimicrobial activity against mature *S. aureus* biofilms. Mature biofilms can play an important role in the persistence of chronic SA infections by decreasing the susceptibility of microbes to antimicrobials by impairing the host immune response [127]. Impairment, specifically phagocytic (macrophage) impairment, can extend the host’s infection length and recovery time. Consequently, there is mounting focus on immunogene therapy to augment the immune system’s initial response. A potential immunotherapy to alleviate macrophage impairment was employed by Kim et al., in which fusogenic liposomes as part of a (small interfering RNA) siRNA–SiNP delivery platform were utilized to bypass the cellular endocytosis’s primary uptake pathway, achieving potent gene knockdown efficacy (Figure 4) [128]. Results showed that the said platform enhanced macrophages’ clearance capability and survivability in a SA pneumonia mouse model. In addition, Liu et al. also used liposomal delivery of antisense siRNA for *mecA* knockdown to restore MRSA susceptibility to oxacillin under both in vitro and in vivo conditions [129].

As well as pH gradients, a-toxin (alpha-hemolysin) (Hla) has also been used to trigger localized drug release from phosphatidylcholine cholesterol-rich liposomes [130,131]. Alternatively, Gram-positive peptidoglycan-specific lysostaphin (LV) can be employed. Recent studies by Hajiahmadi et al. [132] explored the antibacterial activity of vancomycin (free Van) and lysostaphin (free Lys), and lysostaphin–vancomycin (lys/van), liposomal vancomycin (LV), lysostaphin-conjugated liposomes without vancomycin (LysL), and lysostaphin-conjugated liposomal vancomycin (LysLV) against MRSA and *S. aureus*. The authors reported that LV and Van had a similar antibacterial effect against MRSA, whereas the MIC value for free Lys was lower than LysL. In addition, in vivo and MRSA mortality murine studies showed LysLV was the most effective, followed by free Lys/Van, with LysLV significantly reducing the number of bacteria in the surgical site compared with other formulations at the end of the 9th and 14th days.

In addition to glycopeptides, other antibiotic classes have efficaciously benefited from liposomal encapsulation, many of which are addressed in numerous reviews [133,134]. The narrow-spectrum antibiotic Dicloxacillin (DLX) is particularly noteworthy as it has significant activity against Gram-positive β-lactamase-producing microorganisms. In a recent study, researchers [135] prepared a dicloxacillin-loaded liposome using a lipid film hydration method and a chitosan-coated dicloxacillin-loaded liposome via an electrostatic deposition method. Particle sizes of both liposomal formulations were in the nano range (178.5 ± 13.6 nm for DLX-liposomes and 263.4 ± 19.1 nm for chitosan-coated DLX-liposomes). In addition, DLX encapsulation was higher in the chitosan-coated liposomes than the uncoated-liposomes, with encapsulation values of 62% and 38%, respectively. Chitosan-coated and uncoated liposomal formulations exhibited enhanced anti-MRSA activity (inhibition zone of 33.0 ± 0.89 mm for free DLX; 34.3 ± 0.51 mm for chitosan-coated liposomes; and 55.0 ± 1.70 mm for DLX-liposomes), compared to the free drug. These liposomes are believed to show promising potential for their application as a delivery system for DLX, subject to extensive validation studies.

Conventional liposomes (CLs), deformable liposomes (DLs), propylene glycol-containing liposomes (PGLs), and cationic liposomes (CATLs) encapsulating azithromycin (AZT) represents a promising approach for the efficient topical treatment of skin infections. In a study by Vanic et al. [136], AZT encapsulated in CATLs, DLs, and PGLs liposomes resulted in markedly improved in vitro antibacterial activity against planktonic bacteria compared to (aq) free AZT. In addition, these liposomes were superior to free AZT in preventing biofilm formation, exhibiting MIC and minimal biofilm inhibitory concentrations up to 32-fold lower than those of AZT solution, thus confirming their potential for improved topical treatment of MRSA-caused skin infections.

### 4.3. Polymeric Nanoparticles

Chitosan (CS) is a natural biopolymer obtained from one of the most abundant polysaccharides in nature, chitin. CS nanoparticles have been used in oral, nasal, mucosal, ocular, pulmonary, and gene–drug delivery platforms [137]. Positively charged chitosan exhibits good antibacterial activity and the ability to re-potentiate antibiotics [138]. For example, Jamil et al. utilized CS to synergistically enhance the bactericidal activity of β-lactam antibiotics against MRSA biofilms [139]. Chitosan may also improve the applicable lifetime of antimicrobial essential oils (EOs), such as curcuminoids [140,141,142] and cardamom. For example, researchers recently prepared cardamom oil–chitosan nanoparticles by the ionic gel method, demonstrating an encapsulation rate greater than 90%, biocompatibility, and antibacterial activity against MRSA [143].

Approved for a multitude of biomedical applications by the FDA, bovine serum albumin (BSA)-stabilized poly (lactide-co-glycolide acid) (PLGA) exhibits excellent biocompatibility, non-toxicity, and low immunogenicity [144]. Furthermore, the versatility of PLGAs NPs has been successfully utilized in the targeted delivery of antibacterial and anti-inflammatory agents in a sepsis model [145]. The resistance of MRSA primarily lies in its ability to reduce the uptake of free antibiotics and enhance drug efflux. Thiyagarajan et al. sought to circumvent these hurdles by developing a pyridinium amphophilic PLGA nanoparticle system (C1-PNPs) loaded with either gentamicin or ciprofloxacin [146]. Deployment of this combined system restored the susceptibility of MRSA to the antibiotics since C1-PNPs enhanced the cell uptake of gentamicin by MRSA and inhibited the efflux mechanism of MRSA for ciprofloxacin; the authors also postulated that the system has the potential to restore the phagocytic activity of MRSA-infected macrophages.

Similarly, Pei et al. developed a PLGA-based functional nanosystem consisting of PEG-PLGA, Eudragit E100, and a chitosan derivative for intracellular delivery of vancomycin [147]. They found that the nanosystem (500–1000 nm) exhibited increased release at acidic pH and significantly higher uptake levels and MRSA clearance in infected macrophages compared to the control. More recently, Cabral et al. [148] investigated the antibacterial potential of conjugated holo-transferrin (h-Tf) VM-loaded PLGA-PVA nanoparticles against MRSA. Unfortunately, bioconjugation with h-Tf did not increase the antimicrobial effect compared to the unconjugated control. However, the authors did suggest further investigations involving MRSA films and the h-Tf conjugate would be more fruitful.

### 4.4. Solid Lipid Nanoparticles

Solid lipid nanoparticles (SLNPs), also known as lipid carriers, have been under intensive research over the past decade. SLNPs are extensively studied worldwide and have demonstrated significant promise when delivering anti-MRSA antibiotics. Solid lipid nanoparticles (SLNP) consist of solid lipids, surfactants, and co-surfactants. Compared to most other lipid-based nanocarriers (liposomes), SLN remains in the solid state after administration, making them more stable in the gastrointestinal GI environment, shielding cargo (protein and drugs) from enzyme degradation [149]. SNLPs are fabricated from a blend of solid lipids or wax, resulting in a lipid core at room and body temperature. The size and physicochemical properties of SLNPs are readily tunable, depending on the lipids and surfactants used. SNLPs have been shown to act as carriers for hydrophilic vancomycin by ion-pairing the drug with triethylamine and a lipophilic contra-ion (linoleic acid). Sonawane et al. rendered vancomycin SNLPs pH-responsive using a stearic acid-based, cleavable lipid [150]. These site-specific targeting particles gave a 22-fold improvement in MRSA clearance in a mouse skin infection model compared to the controls. More recently [151], researchers utilized an N-(2-morpholinoethyl) oleamide (NMEO) pH-responsive lipid for vancomycin delivery and examined its stability and antibacterial activity in neutral and acidic pH. The study revealed that drug release and antibacterial activity were significantly better at pH 6.0 than pH 7.4. Moreover, the MRSA load was 4.14 times lower (*p* < 0.05) in Van NMEO SLNPs treated mice than bare VM-treated specimens. Govender et al. also demonstrated the improved efficacy of Van delivered via novel oleylamine-based zwitterionic lipid (OLA), chitosan-based, pH-responsive lipid–polymer hybrid nanovesicles (Van-OLA-LPHVs1) in the treatment of MRSA [152]. Van release from the Van-OLA-LPHVs1 was faster at pH 6.0 than pH 7.4, with 97% release after 72 h. The Van-OLA-LPHVs1 had a lower MIC value of 0.59 μg/mL at pH 6.0 compared to 2.39 μg/mL at pH 7.4 and a 52.9-fold antibacterial enhancement compared to the control. In vivo studies in a BALB/c mouse-infected skin model treated with Van-OLA-LPHVs1 revealed a 95-fold lower MRSA burden than the bare Van group. The same group also [153] addressed the problem of intracellular infection by developing novel pH-responsive lipid–dendrimer hybrid nanoparticles (LDH-NPs) for the intracellular delivery of vancomycin. Bacterial cell viability studies showed that LDH-NPs killed 84.19% of the MRSA, compared to Van (49.26%) at the same MIC, confirming its enhanced efficacy. Cell uptake studies showed that LDH-NPs intracellularly accumulated in HEK 293 cells, demonstrating significant clearance (*p* < 0.0001) of intracellular bacteria.

A more direct method in inhibiting MRSA growth involves the employment of transcription factor decoys (TFDs). TFDs are short-length oligonucleotides (10–80 base pairs) carrying a bacterial essential transcription factor [154]. When a bacterial cell is transformed with these molecules, the TFDs outnumber the native promoter binding sites in the chromosome [155]. However, the efficient intracellular delivery of the TFDs is critical in realizing the antibacterial potential of this technology. Initial studies utilizing specific TFDs complexed with either cationic nanostructured lipid carriers (cNLCs) or chitosan-based nanoparticles (CS-NCs) found that both carriers were adept at complexing and protecting TFDs in a range of physiological and microbiological buffers up to 72 h. Initials tests showed that the “anionically” charged chitosan-TFD particles demonstrated no visible improvements in effect when co-administered with vancomycin. However, co-delivery of cNLC-TFD with vancomycin reduced the MIC of vancomycin by over 50% in MSSA and resulted in significant decreases in viability compared with vancomycin alone in MRSA cultures. Optimizations of the nanocarrier composition and the sequence and structure of the TFD molecule are being carried out to improve their combined efficacy against MRSA.

Plants contain rich sources of bioactive phytochemical compounds that exhibit broad-spectrum antibacterial activity; 18β-glycyrrhetinic acid is such a compound [156]. In a recent study [157], the targeting capability of pH-responsive lipid(oleic)-polymer hybrid nanoparticles (LPHNPs) was employed in the co-delivery and enhancement of the antibacterial activity of vancomycin and 18β-glycyrrhetinic acid. By co-encapsulating Van and 18β-glycyrrhetinic acid within LPHNPs, their pharmacokinetic profiles and therapeutic indices were remarkably enhanced. Moreover, studies revealed that LPHNPs loaded with 18β-glycyrrhetinic acid and Van exhibited sustained and faster release in acidic conditions and a 16-fold increase in antibacterial activity against MRSA compared to bare Van suggesting encapsulated Van and 18β-glycyrrhetinic acid acted synergistically. Given that 18β-Glycyrrhetinic acid (GA) has the ability to regulate the production of haemolysins, leukotoxins, and adhesins [158,159], it would seem that this platform has the potential to modulate virulence as well. Furthermore, 18β-glycyrrhetinic acid is readily available, suggesting this platform represents a cost-effective, non-toxic treatment option for MRSA. A compilation of the various carriers encapsulating vancomycin used in the treatment of MRSA is shown in Table 2.

### 4.5. Stealth Coatings (Delivery and Detoxification)

PEGylated liposomes, LNPs, and other lipid-based drug delivery systems (DDS) were originally thought to be immunologically inert. However, repeated administration of PEG-nanoparticles resulted in the production of antibodies (IgM and IgG) against carrier components resulting in infusion reactions such as complement (C) activation-related pseudo allergy (CARPA) [160]. CARPA may be perceived as an immunological response to structural similarities common to nanomedicines and viruses [161]. The entailing acute inflammatory reaction may result in reduced efficacy, anaphylaxis, and immunogenicity (antibody generation) [162,163]. Alternatives to PEG, such as polyglycerol [164], are beyond the scope of this review. For those readers interested in naïve PEG antibodies, their prevalence within the general populous and the potential impact on therapeutics, the review by Hong et al. is recommended [165].

In contrast to PEGs’ susceptibilities to clearance, alternative coatings, such as erythrocyte membrane and platelets, have been used to extend the circulatory lifetime of (PLGA) (NPs), perfluorocarbons (PFCs)–PLGA nanoparticles, up-conversion nanoparticles, and metal–organic frameworks (MOFs) [166,167,168]. Recently, Huang et al. [169] examined the antibacterial potential of platelet encapsulated Ag-MOF loaded with vancomycin (PLT@Ag-MOF-Vanc) against *S. aureus* and MRSA. PLT@Ag-MOF-Vanc showed better antibacterial activity against MRSA in vitro than free vancomycin and Ag-MOF, Ag-MOF-Vanc groups (Figure 5). In addition, the carrier exhibited targeted release, killing MRSA through multiple approaches, including interfering with the metabolism of bacteria, catalyzing reactive oxygen species production, destroying cell membrane integrity, and inhibiting biofilm formation. Moreover, PLT@Ag-MOF-Vanc demonstrated reduced phagocytic uptake compared to the controls (Ag-MOF, Ag-MOF-Vanc groups and vancomycin group). Furthermore, the study also evaluated the anti-infection effect of PLT@Ag-MOF-Van in an MRSA pneumonia model of Kunming mice. The results showed better and faster recovery in the lung condition in the PLT@Ag-MOF-Vanc group compared with other groups, and the alveoli recovered from the third day of the treatment, with no apparent inflammatory cell infiltration.

Rich in complement and transmembrane proteins such as CD47 [170], CD59 [171], decay-accelerating factor (DAF) [172] and complement receptor 1(CR-CD35) [173], flexible RBC membranes have been shown to delay the opsonization of nanoparticles for several months. This, in turn, has allowed researchers to explore the encapsulation of vancomycin by RBC membrane-derived vesicles supplemented with exogenous cholesterol [174]. Van-RBC nanoformulations demonstrated higher retention at MRSA-induced infection sites in murine models and reduced skin lesion formations. In addition, bacterial enumeration revealed that Van-RBC could outperform the free drug by three orders of magnitude.

The deadly nature of *S. aureus* is attributable to the release of bacterial toxins, including α-, β-, γ-, and δ-pore-forming toxins, exfoliatin, enterotoxins that cause toxic shock and scalded skin syndrome, and poisoning from infected food. In addition, many of these pore-forming toxins activate intracellular K+ sensors, leading to a pathway that modifies histones and subsequent gene expression, predisposing the host to recurring and secondary infections [175,176]. An insightful approach employed by Zhang et al. to accelerate the removal of these toxins was to combine the capturing capacity of erythrocyte membranes with freshly prepared vancomycin nanosponges (NS) in the treatment of MRSA infections [177]. Compared with free Van and nonresponsive nanogels, the coated nanogels exhibited remarkable antibacterial activity. Furthermore, researchers demonstrated the intracellular antibacterial efficacy of vancomycin-loaded RBC-nanogel in an in vitro model of MRSA USA300-infected macrophages sourced from human THP-1 monocytes. However, nanogels prepared using the cross-linker N, N, N′, N′-Tetramethylethylenediamine (TEMED) and catalyst ammonium persulfate (APS) [177] may compromise biocompatibility and limit its translational applicability.

An alternative approach is to employ “smart” thermosensitive hydrogels based on Pluronic F127 (an FDA-approved novel temperature-sensitive hydrogel material) that rely on physical methods for cross-linking [178]. In a recent study, Zhang et al. [179] successfully used RBC-derived nanosponges and the FDA-approved Pluronic F127 hydrogel to construct a novel biocompatible, biodegradable detoxification system denoted as “NS-pGel”. NS-pGel was shown to preserve the Hlα neutralization capability of the incorporated NSs and significantly prolonged retention of NSs in both biological buffers and mouse subcutaneous tissues. Moreover, the prophylactic detoxification potential of NS-pGel showed better preventive effects than NSs alone.

In addition to detoxification, eukaryotic or prokaryotic sourced EVs have been used to enhance the immunogenetic or therapeutic effects for preventing and treating bacterial infections. For example, *S. aureus* EV-coated magnetic mesoporous silica loaded with indocyanine green triggered multi-antigenic vaccination and modulated antigen presentation pathways to activate T cells responses [180]. In another study [181], *S. aureus* EVs were utilized to coat poly (lactide-co-glycolide acid) (PLGA) nanoparticles preloaded with antibiotics. Due to their antigenic properties, the EV-coated nanoparticles were effectively internalized by *S. aureus*-infected macrophages and released antibiotics to kill the intercellular pathogens, offering significantly improved efficacy in alleviating *S. aureus* burdens.

## 5. Biofilms

A potential consequence of the pandemic [182] is a rise in the frequency of biocide resistance genes qacA/B and qacC in clinical staphylococci isolates, particularly MRSA [183], which may enhance antibiotic cross-resistance within the broader community. A typical example of cross-resistance (CR) is that of (quaternary ammonium compounds) “QAC” transporters, which enhance the efflux of clinically relevant antibiotics [184], particularly aminoglycosides. CR can occur in environments where poor or inexperienced sanitation practices are adopted, leading to resident microbes being exposed to sub-lethal concentrations of biocides. Moreover, Pereira et al. [185] recently demonstrated that the evolution of 40 Escherichia coli strains in sub-inhibitory concentrations of 10 (including chlorhexidine) widespread biocides resulted in 17 strains exhibiting reduced susceptibility to medically relevant antibiotics. In addition, 11 of those strains showed a greater capacity for biofilm formation. Perhaps more concerning were the studies by Durna and Speck et al. [186,187], which showed sub-MICs of sodium hypochlorite enhanced the biofilm-forming ability of MRSA and increased resistance to oxacillin in *Staphylococcus aureus* after exposure to sub-lethal sodium hypochlorite concentrations.

Greater capacity for MRSA biofilm generation lies in the upregulation of pro-biofilm genes such as *fnb*, *agr*, *sarA*, and *icaADBC* [188]. The ability of MRSA to colonise and persist (as biofilm) on implants [189,190] (orthopaedic, heart valves, and shunts) and medical devices, such as catheters, endotracheal tubes [191], and pacemakers, are well known. MRSA infections can be chronic and recurrent. In addition, the pathogen can colonize virtually any biological or inanimate surface and has been identified in industrial and domestic settings [186]. Biofilm formation occurs in four stages [192]: planktonic cell adhesion to a substrate; early micro-colony proliferation and polysaccharide intercellular adhesion (PIA) production; secretion of extracellular eDNA (biofilm maturation); and surfactant-aided detachment of bacteria. The primary oligosaccharide in SA biofilm matrices is a polymer of N-acetyl-β-(1-6)-glucosamine (polysaccharide intercellular adhesin or PIA), and accumulation-associated protein (Aap), a common biofilm-associated protein [193]. The characteristic features of a biofilm that afford it resistance to biocides and antibiotics alike are depicted in Figure 6. For a more extensive review on alternative strategies used in biofilm elimination, the study by Koo and colleagues is recommended [194].

Since biofilms usually house multispecies and are 100–1000 times more antibiotic-resistant than their planktonic counterparts (and other microorganisms), attempts to eliminate these diverse bacterial communities with high-dose, single antibiotics can result in toxicological damage to the host. Consequently, many researchers have focused on developing inhibitory strategies. Such strategies have included the coating or doping of surfaces (implants and devices) with antibacterial agents such as antibiotics [195], silver nanoparticles [196], MgB2-polyvinylpyrrolidone (PVP) composites [197], antimicrobial peptide (AMP) [198], and F-18 bio-glass [199,200]. Antibiotics can be tethered to the surface of an implant or incorporated as a part of a nanocomposite scaffold. In a recent study [201], the antibacterial activity of a gelatin–strontium-incorporated hydroxyapatite (SrHAP)-forming HG scaffold and vancomycin-loaded chitosan–gelatin polyelectrolyte complex-incorporated gelatin-SrHAP-forming HV scaffold (HV1–0.5 wt% and HV2–1 wt% vancomycin) were investigated. The HV-2 sample showed significant antibacterial activity for MRSA and MSSA compared to HV1 and the controls. A more conventional approach is to coat the base material with Ag, Cu, Zn, Au, and Ni particulates. The antibacterial mechanism of AgNPs via Ag^+^ (ROS elevation) release on planktonic microbes is well known [202]. Moreover, silver nanoparticles have broad-spectrum appeal killing both Gram-negative and Gram-positive bacteria alike [202]. Currently, the biggest challenge facing AgNPs is sustained ion release. To meet this challenge, researchers have used Ti nanotubes loaded with polydopamine (PDA)-coated Ag_2_O NPs. Investigations showed long term improvements in sustained release and reduced host toxicity compared with uncoated AgNPs [203]. Similarly, TiO_2_ nanorods and AgNPs were used by Guan et al. [204] to measure the antibacterial coating efficacy of Ag-TiO_2_@PDA in a series of in vitro experiments. Experiments showed that Ag-TiO_2_@PDA NRDs coatings demonstrated controlled Ag^+^ release with anti-MRSA effects on Days 7 and 14, exhibiting efficiencies of 88.6 ± 1.5% and 80.1 ± 1.1%, respectively. The anti-MRSA activity of Ag^+^ was confirmed in-vivo following implantation in the tibia of an osteomyelitis rat model. Aside from implants, for the past 30 years, silver-coated medical devices have been intensely investigated [205]. Several studies have shown that silver-coated endotracheal tubes can reduce the occurrence of early-onset ventilator-associated pneumonia by preventing biofilm formation [206,207]. Silver nanoparticle-based antimicrobials can promote a long-lasting bactericidal effect without detrimental toxic side effects. However, translation to the clinical settings remain slow as no clear and complete protocol defines the particles’ specific toxicity (size, shape, surface charge, and ionic content), restricting clinical application [208].

In addition to silver-based nanoparticulate coatings, other elements composed of copper have been used to prevent biofilm formation. The antibacterial ability of Cu largely depends on its form (ion or nanoparticle), oxidation state (Cu^0^, Cu^1+^, or Cu^2+^), and concentration. In addition, the contact distance between microorganisms and Cu-containing surfaces, application form (dry or wet), and ambient temperature significantly affect its antibacterial potential [209]. For those readers wanting to explore current research on antibacterial metals and alloys used in implants, the reviews by Jiao [210] and Liu et al. are recommended [211].

## 6. Clinical Translations

Nanoparticulate pharmaceutical drug delivery systems (NDDSs) are commonly used to increase the efficacy of medicines. However, less than 60 drug-loaded nanoparticles have been approved for commercial use [212,213,214]. Cancers naturally dominate the treatment landscape, followed by blood disorders, chronic diseases, and fungal infections. Regarding MRSA [215], following phase I trials, the latest Egyptian observational study involving 150 patients (ClinicalTrials.gov Identifier: NCT04431440), using topical silver nanoparticles, has shown promising results; the stability data are yet to be published. Other trials (NCT04775238) involving copper and silver nanoparticles synthesized using laboratory procedures are still recruiting patients.

Several criteria have to be met for a nanoparticle formulation to succeed in the clinic. These include reliably scaling up synthesis, high throughput optimization, and predicting nanoparticle efficacy and performance. Small batches of nanoparticles sourced from a conventional laboratory usually suffer from a high degree of variability in size and toxicity, preventing scaling up synthesis by manufacturers. High-throughput nanoparticle optimization involves large-scale screening of numerous formulations performed preclinically for specific biological functions or in vitro release profiles, utilizing selective iterations, leading to a single specific function. Unfortunately, this technology remains in its infancy; thus, a strong correlation between human and animal models is still relied upon at the preclinical stage, despite the issue of nephrotoxicity. To further optimize nanoparticle performance, individual taxon-based gut analysis before a study could provide an additional level of specificity, as there is a strong preclinical and clinical rationale (doxorubicin-metabolite) to incorporate this iteration [216]. However, this would require metagenomic information sharing between individuals, biotech (microbiome), and drug companies. In the future, one could envision microbial host–animal surrogate models with virome specificity, although without governmental assistance, this may prove economically unfeasible for the public at large.

## 7. Conclusions

The MRSA arsenal of toxins, resistant genes, and adhesins [217] represents a unique set of challenges in terms of vaccine development, diagnostics, treatment, and biofilm inhibition. Multiple MRSA vaccine trials have failed to meet their endpoints, whilst EV-based MRSA/SA vaccination platforms remain in their infancy. Thus, the potential to reduce the spread of multi-drug resistant SA and antibiotic usage via vaccination is currently unavailable. However, EV versatility represents a unique opportunity for the development of novel toxins, vaccines, stealth, and antibiotic carriers [218]. Conversely, in antibiotically challenged bacterial hands, EVs can act as membrane decoys, carrying lactamases into the microenvironment whilst enabling the transfer of AMR genes to susceptible bacteria. Monitoring or sensing EV production would magnify our understanding of AMR gene transfer and host toxicity issues stemming from free-form antibiotics. In this regard, nanomaterials may play a pivotal role in MRSA EV diagnostics and infection control [219].

Nanomaterials (metallic, polymer, and liposomal) have proven to increase the sensitivity of a wide variety of optical and electrochemical MRSA bacterial diagnostic assays and sensors. The application of these materials is so prevalent that NP-based colorimetric lateral flow assays can now be conducted in remote, low-resource settings, with results relayed instantaneously via mobile applications to city medical centres. In addition, further studies utilizing these materials may allow for the effective monitoring of potential surges in MRSA and VRSA infections in isolated regions where the risk of an outbreak is the strongest. Yet however simple a POC device is, training in the field or a doctor’s surgery is required, which in its absence may hinder compliance. In addition, a field device might need to be more robust and resistant to contaminants and interferents (increase costs) than those employed in the clinic. Finally, the question of extra information provided by enhanced sensitivity imparted by nanomaterials and clinical relevance. Extra information is required in recurrent/sepsis (MRSA) infections and pandemics in which the patient’s condition can rapidly change. For those readers interested in the translation of POC devices into health care, the review by Dhawan et al. is recommended [220]

Given eukaryotic toxicity issues surrounding the administration of free-form Van, multiple attempts to improve its efficacy via encapsulation (EE > 50%) using a variety of nanomaterials, including pH-responsive fusogenic liposomes and SLNPs, have resulted in improvements in efficacies by more than an order magnitude. Moreover, these carriers would be ideally suited for the delivery of Van adjuvants(b-lactams) as well. However, the usage of antibacterial nanoparticles such as silver and gold often fails to meet clinical requirements, leading to agglomerates in peripheral tissues, limiting their application to antibacterial coatings and topical applications.

In addition to pH targeting and adjuvants, a carrier that delivers a toxin modulator could limit infection severity. To this end, researchers encapsulated Van and 18β-Glycyrrhetinic acid (GA) (a known toxic regulator) using a pH-responsive carrier, resulting in more than a magnitude increase in Van efficacy and a significant reduction in hemolysin production, representing a non-toxic, cost-effective treatment option for MRSA in low-resource settings.

## Figures and Tables

**Figure 1 pharmaceutics-14-00805-f001:**
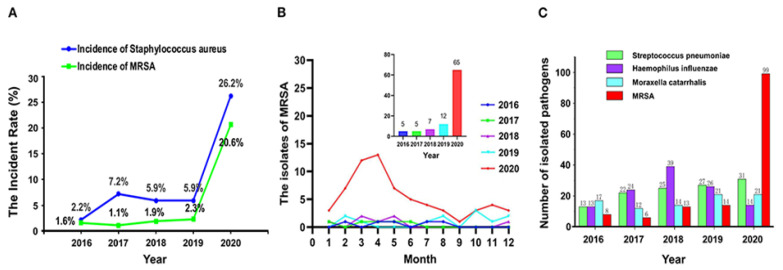
The detection rate of *Staphylococcus aureus* and methicillin-resistant *S. aureus* (MRSA) (**A**), the isolates of MRSA (**B**), and the number of pathogenic bacteria (**C**) isolated from respiratory specimens from 2016 to 2020. Reproduced and modified with permission [13] (2021).

**Figure 2 pharmaceutics-14-00805-f002:**
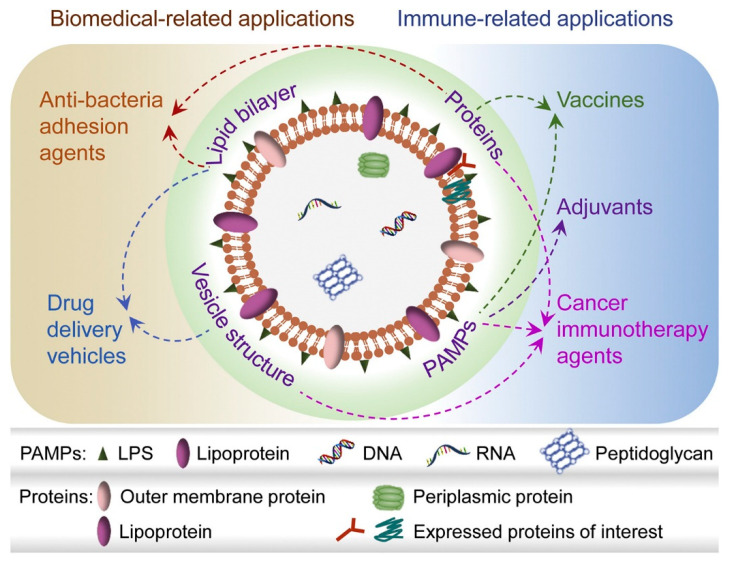
Composition of natural and genetically engineered OMVs, and contributions to biomedical applications. (1) OMVs with multiple PAMPS enhances antigen-specific immune responses; additional adjuvants are not required. (2) OMVs can be tailored with foreign proteins/polypeptides. (3) The vesicular structure, PAMPs, and proteins target tumour and infection sites and elicit a robust immune response. (4) The vesicle structure of OMVs formed by lipid bilayers permits carriage of drug, gene, or protein cargos (5). Anti-adhesion agents allow OMVs to complete with toxin-secreting pathogens. Reproduced and modified with permission from [39] Copyright (2020) Elsevier.

**Figure 3 pharmaceutics-14-00805-f003:**
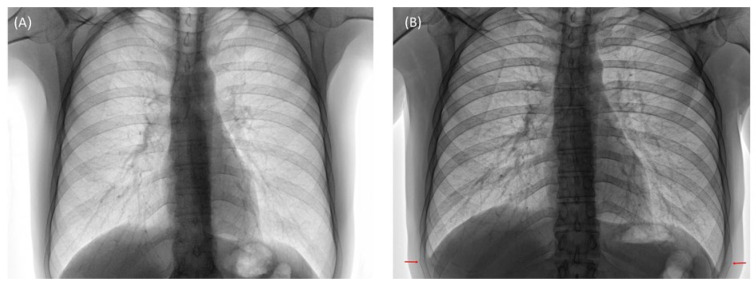
Computed tomography (CT) posterior–anterior lung radiographs of patient X. (**A**) Before flu infection; Day 1: 39.5 °C evening after school trip, sweating. Days 2–4: 38.5 °C violent coughing, Days 5–8: 38 °C violent coughing, and *S. aureus* secondary-infection(sputum) Day 9, 10: 37 °C sudden sharp pains. (**B**) Two weeks, resultant bilateral fracture (ninth ribs) as highlighted.

**Figure 4 pharmaceutics-14-00805-f004:**
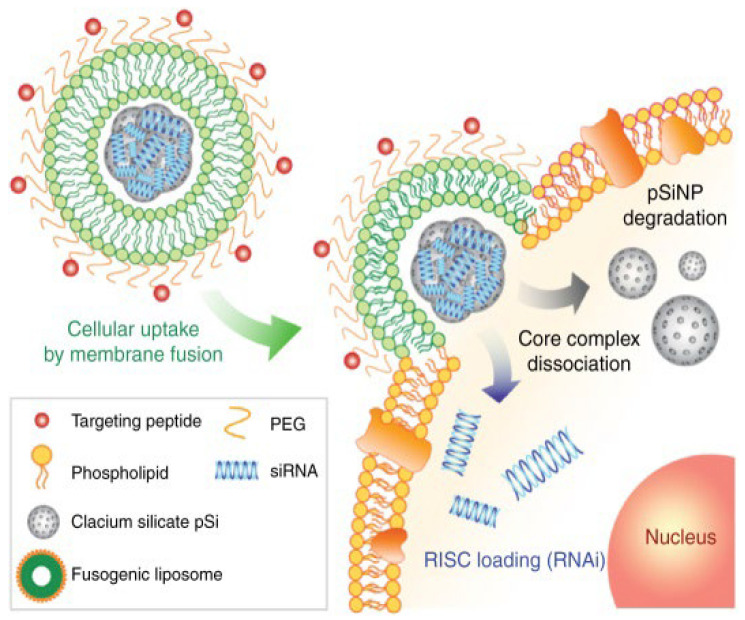
Schematic depicting the siRNA cargo’s fusion, core internalization, and cytosolic dissociation within a macrophage. Reproduced with permission from [128]. Copyright, (2018) Springer Nature.

**Figure 5 pharmaceutics-14-00805-f005:**
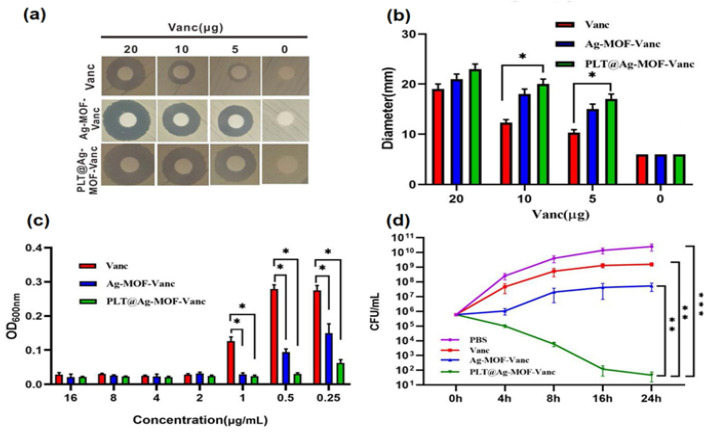
In vitro antibacterial effect of Ag-MOF-Vanc: (**a**) Inhibition zones, (**b**) corresponding inhibition zone diameters, and (**c**) concentration effects of Vanc, Ag-MOF-Vanc, and PLT@Ag-MOF-Vanc against MRSA. (**d**) CFU of MRSA treated with 0.5 μg/mL of different drugs. Data are presented as the means ± SD (*n* = 3). * *p* < 0.05, ** *p* < 0.01, and *** *p* < 0.001. Reproduced with permission [169]. Copyright (2019) Springer Nature.

**Figure 6 pharmaceutics-14-00805-f006:**
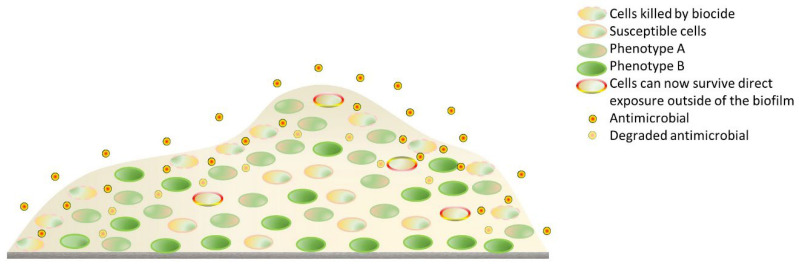
The dense extracellular biofilm can degrade and restrict the penetration of biocides and antibiotics alike. In addition, bacteria can acquire cross-resistance to an antibiotic after sublethal exposure to biocides.

**Table 1 pharmaceutics-14-00805-t001:** Various techniques utilizing nanomaterials in the detection of MRSA.

Technique	Nanoparticulate	LOD	Detection Target	Assay Time	Ref.
SERS	Ag NPs	10 CFU/mL	MRSA	/	[64]
SERS	AgNPs+	/	MRSA	45 min	[67]
Colourimetric PCR	AuNPs	500 ng	*mecA*	<25 min	[70]
Colourimetric	AuNPs	100 ng	*mecA*	<100 min	[71]
Colourimetric	AuNPs	500 ng DNA	*mecA*	<60 min	[73]
Resistive pulse sensing	AuNPs	530 copies	*PVL* gene	120 min	[77]
Fluorescence	GO	0.02 nM	MRSA 16S rRNA	/	[90]
LRET	UCNs	0.18 nM	mec-Tar	/	[91]
FRET	CdTe QD	0.5 ng/mL	Antibodies	/	[92]

**Table 2 pharmaceutics-14-00805-t002:** Comparison of in vitro anti-MRSA activity of encapsulated free vancomycin.

Carrier	Cargo	MIC	MIC Free Form	Ref.
Liposome	Van	0.3 mg/L	1.25 mg/L	[119]
Liposome	Van	0.48 µg/mL	7.68 µg/mL	[124]
OLA-LPHVs	Van	0.59 µg/mL	31.25 μg/mL	[152]
LDH-NPs	Van	3.90 μg/mL	31.25 mg/mL	[154]
LPHNPs	Van & 18β-glycyrrhetinic acid	0.48 μg/mL	7.81 mg/mL	[157]

## Data Availability

Not applicable.

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
