# Peer review of "Application of Nanomaterials in the Prevention, Detection, and Treatment of Methicillin-Resistant Staphylococcus aureus (MRSA)"

_pharmaceutics, 2022, doi:10.3390/pharmaceutics14040805_

Round 1
Reviewer 1 Report
The paper “Application of nanomaterials in the prevention, detection, and treatment of Methicillin-resistant Staphylococcus (MRSA)” by John Hulme reviews some important aspects of a major problem, the antimicrobial resistance. The review is well organized and the author summarize five possible ways to combat MRSA. The tables and figures are very comprehensive.
The article has been read over and is worthy of publication.
Author Response
Thank you
Reviewer 2 Report
The aim of this review is to discuss the current potential of the nanomaterials in detecting and treating MRSA. The review discusses five areas where natural and synthetic delivery carriers/vehicles are used to combat MRSA such as vaccines, rapid diagnostics, antibiotic delivery, nano-stealth coatings and biofilm inhibition.
The review is well-presented and the topics discussed are interesting, however the issues listed below need to be addressed before re-submission of the manuscript.
Major issues:
- The Antimicrobial agents section needs to be revised. The separate subsection which will include Ag, Au, metal oxide NPs should be added. The introduction paragraph following subsections should be also included to introduce the readers to the following subsections.
- In the antimicrobial agents section the authors included silver nanoparticles (AgNPs), which are found that directly involved in mitochondrial toxicity and DNA damage. Therefore, there is concerns of using AgNPs for various health care applications. The authors should make some comments in regard the safety of using AgNPs and their cytotoxicity for biomedical applications.
Minor issues:
- The bacteria strain names should be presented in Italic.
Author Response
The Antimicrobial agents section needs to be revised. The separate subsection which will include Ag, Au, metal oxide NPs should be added. The introduction paragraph following subsections should be also included to introduce the readers to the following subsections.
The separate subsection created 4.1 Metallic nanoparticles
Text expanded
Extra 4 references added
In the antimicrobial agents section the authors included silver nanoparticles (AgNPs), which are found that directly involved in mitochondrial toxicity and DNA damage. Therefore, there is concerns of using AgNPs for various health care applications. The authors should make some comments in regard the safety of using AgNPs and their cytotoxicity for biomedical applications.
Discussed at length lines 384-391
Italic issues fixed
Thank you.
Reviewer 3 Report
This is a well-organized and well-illustrated review article, with an important clinical message for the prevention and treatment of antimicrobial diseases, in particular Methicillin-resistant Staphylococcus (MRSA), and should be of great interest to the readers. The review focused on the recent developments in the application of nanomaterials for the prevention, detection, and treatment of MRSA. Paragraphing is concise and good, and the article consists of major recent advancements in the field of Applied nano systems for antimicrobial therapies and detection. This review article deserves publication after some revisions listed below.
- In lines 47, increased frequency of disinfection is known to cause the higher incidence of MRSA, what is the authors opinion or advice to overcome this shortcoming?
- The introduction section describes about the challenges of MRSA and drug resistance but however, I suggest the authors to at least include a brief paragraph on how the nanoparticle-based therapies can be advantageous over the currently existing therapies.
- I suggest the authors to write a brief section about the limitations of currently available nanomaterial-based technology for clinical translation and what measured must be followed for their successful clinical translation?
- It would be better if the authors can tabulate the nanomedicines for antibacterial use or detection that are in clinical trials or passed clinical trials.
- The questions that author missed to address in conclusion:
There is always a dilemma on how to conclude a review article. Since the authors have deliberately summarized huge amounts of published results, it will go a long way. It would be helpful if they can provide their own thoughts that would in turn help in finding the areas that need to be addressed. For example, what are the factors that one needs to consider while choosing an ideal nanocarrier for MRSA detection or anti-microbial drug delivery, what are the required criteria to overcome the toxicity associated with each of these formulations and what are the steps required for the fast transition of these materials for industrial scale up. In general, what measures need to be taken for the effective clinical translation of anti-microbial or diagnostic nanocarriers? Though lipid based nanoparticles like liposomes are studied from the past two decades, why there are only a handful of FDA approved anti-microbial formulation or diagnostic kits? What limitations are hindering their clinical translation and in what direction does the future research need to be, to make the clinical translation possible?
Author Response
Points 1-5 addressed in blue
For the majority, we concur with referee 4.5

Round 2
Reviewer 2 Report
The authors amended the manuscript as per reviewer's comments. The revised manuscript is acceptable for publication.